# IL-25 blockade augments antiviral immunity during respiratory virus infection

Teresa C. Williams[1], Su-Ling Loo[1], Kristy S. Nichol[1], Andrew T. Reid[1], Punnam C. Veerati [1], Camille Esneau[1], Peter A. B. Wark [1,2], Christopher L. Grainge[1,2], Darryl A. Knight[1,3,4], Thomas Vincent[5], Crystal L. Jackson[5], Kirby Alton[5], Richard A. Shimkets[5], Jason L. Girkin [1] & Nathan W. Bartlett [1✉]

IL-25 is implicated in the pathogenesis of viral asthma exacerbations. However, the effect of IL-25 on antiviral immunity has yet to be elucidated. We observed abundant expression and colocalization of IL-25 and IL-25 receptor at the apical surface of uninfected airway epithelial cells and rhinovirus infection increased IL-25 expression. Analysis of immune transcriptome of rhinovirus-infected differentiated asthmatic bronchial epithelial cells (BECs) treated with an anti-IL-25 monoclonal antibody (LNR125) revealed a re-calibrated response defined by increased type I/III IFN and reduced expression of type-2 immune genes CCL26, IL1RL1 and IL-25 receptor. LNR125 treatment also increased type I/III IFN expression by coronavirus infected BECs. Exogenous IL-25 treatment increased viral load with suppressed innate immunity. In vivo LNR125 treatment reduced IL-25/type 2 cytokine expression and increased IFN-β expression and reduced lung viral load. We define a new immune-regulatory role for IL-25 that directly inhibits virus induced airway epithelial cell innate anti-viral immunity.

[1] The University of Newcastle and Hunter Medical Research Institute, Newcastle, NSW, Australia. [2] Department of Respiratory and Sleep Medicine, John Hunter Hospital, Newcastle, NSW, Australia. [3] UBC Providence Health Care Research Institute, Vancouver, BC, Canada. [4] Department of Anaesthesiology, Pharmacology and Therapeutics, University of British Columbia, Vancouver, BC, Canada. [5] Abeome Corporation/Lanier Biotherapeutics, Athens, GA, USA.
✉email: Nathan.bartlett@newcastle.edu.au

Virus-induced exacerbations are the leading cause of hospitalization and mortality for individuals with asthma[1]. Rhinovirus (RV) has been detected in 50–70% of patients with asthma who present to the hospital with an exacerbation[2,3]. As a result, the mechanisms by which a viral infection worsens asthma are a major research focus with two prevailing concepts emerging – deficient/delayed innate anti-viral immunity associated with excessive type-2 immune activation causing airway inflammation[4,5].

The role of type-2 immunity has been recognised by the development of biologics (monoclonal antibodies mAbs) that inhibit interleukin 4 (IL-4), IL-5, or IL-13 and reduce the frequency of exacerbations up to 50%[6–12]. More recently, the focus has shifted to airway epithelial cell-expressed cytokines such as TSLP that stimulate type 2 immune pathways[13]. IL-25 is also expressed by epithelial cells and stimulates type-2 inflammation. IL-25 expression is higher at baseline and during RV infection in individuals with asthma[14]. IL-25 signals through an IL-17RA/IL-17RB heterodimer receptor on immune cells, such as type-2 innate lymphoid cells (ILC2), T helper 2 (T$_h$2) cells, eosinophils, basophils, mast cells as well as bronchial epithelial cells (BECs)[15–18], which constitutively express IL-25 for immediate secretion upon exposure to proteases or pathogens[19].

Airway epithelial cells are also the primary site of respiratory viral infection and are critical to initiating antiviral immunity[20]. In the lungs, BECs induce an antiviral response through the production of type I interferon-β (IFN-β) and type III IFN-λ which in turn induce expression of IFN-stimulated genes (ISGs) that directly interfere with viral replication, enhance viral antigen presentation, and activate the adaptive immunity[21,22]. Deficient/delayed type I and type III IFN production by RV-infected BECs from patients with asthma has been identified and this is thought to contribute to enhanced airway inflammation and bronchoconstriction and more severe disease[23–25]. However, the mechanisms underlying inadequate antiviral immunity in asthma and how this contributes to disease are not well understood. We hypothesized that IL-25 directly regulates BEC innate immunity during viral infection and inhibition of IL-25 (in addition to suppressing type 2 inflammation), increases IFN expression and reduces viral load. To define the role of IL-25 in regulating airway-epithelial cells antiviral immunity, we employed an IL-25 monoclonal antibody (LNR125 (generated by Abeome Corporation, now Lanier Biotherapeutics, using the transgenic mouse platform AbeoMouse$^{TM}$), in in vitro and in vivo models of viral infection in asthma. LNR125 upregulated rhinovirus- induced IFN-β and IFN-λ and coronavirus (229E)-induced IFN-λ in differentiated BECs from donors with asthma and healthy donors respectively. Further, LNR125 IL-25 blockade enhanced ISG expression and down-regulated type-2 immune genes. Exogenous IL-25 protein treatment inhibited innate antiviral immunity in RV-infected differentiated human BECs. We used an established mouse model to determine the effect of a single subcutaneous treatment with LNR125 on anti-viral immunity during RV-exacerbation of allergic airways disease[26]. In additional to suppressing inflammation, antibody-mediated IL-25 blockade increased IFN-β expression in airways and reduced lung viral load.

## Results

### RV-A1 infection upregulates IL-25 expression by differentiated bronchial epithelial cells.
We previously reported that RV infection induced higher levels of IL-25 gene and protein expression by undifferentiated (submerged monolayers) BECs from donors with asthma compared to BECs from healthy donors[14]. Here we used human endobronchial biopsies and

differentiated primary human BECs (healthy donors and patients with asthma) to gain insight into coexpression of IL-25 and IL-17RB by differentiated BECs (Fig. 1a, Supplementary Fig. 1a). We first stained the biopsies with haematoxylin and eosin (H&E) to confirm the presence of intact epithelium. We next compared IL-25 and IL-17RB protein expression using immunofluorescence. IL-25 and IL-17RB expression was predominantly located on the apical, luminal mucosal surface of the airway epithelium. We next determined if ALI-differentiated BEC cultures from healthy and asthmatic donors generated for a previous study[25] exhibited a similar IL-25 and IL-17RB expression pattern to that of bronchial biopsies. By comparing the pattern of expression to previously reported histological analyses of these ALI-BEC cultures we again observed that IL-25 and IL-17RB expression was highly localized to the apical surface of BEC cultures from healthy donors and donors with severe asthma with evidence of co-localisation in both (Fig. 1b, Supplementary Fig. 1b). Having determined the ALI-BEC culture system recapitulates in vivo airway mucosal surface expression of IL-25 and IL-17RB, we next determined the effect of RV infection. BECs from $n = 14$ donors with asthma (donor characteristics in Supplementary Table 1) were cultured at ALI for at least 25 days and infected with RV-A1, MOI = 0.1 as previously described[25]. RV infection increased IL-25 gene expression at 2- and 4 days post-infection (Fig. 1c). We next used an in-house-developed sensitive ELISA to investigate secreted IL-25 from apical supernatants of BEC cultures, however, we did not consistently observe increased secreted IL-25 protein in apical media from RV infected BECs. Having observed cell-associated IL-25 in uninfected bronchial biopsies and ALI-BEC cultures by immunofluorescence, we investigated cell-associated IL-25 from BEC cell lysates which revealed RV infection increased cell-associated IL-25 protein on 2 d p.i. (Fig. 1d).

### Antibody blockade of IL-25 augmented RV-induced type I/III IFN expression by BECs from patients with asthma.
Having confirmed that IL-25 was expressed in ALI-differentiated BECs and increased by RV-A1 infection, we next examined if IL-25 was modulating epithelial cell innate anti-viral immunity in a subset ($n = 6$) of BEC from donors with moderate-severe asthma (Supplementary Table 1). Nanostring immune transcriptomics was used to assess the effect of LNR125 IL-25 blockade against the isotype control antibody (LNR2) at 4 d.p.i. in RV infected BECs. Of the 500+ immune genes in the human immune panel, we noted two distinct groups: up-regulated innate immunity-(type I/III IFNs, TLR2, IFR7, TBK-1, IRAK2) and downregulated type 2/immune suppressing-genes (CCL26/eotaxin 3, IL1RL1/ST2, TGFβI) (Fig. 2a, Supplementary Fig. 2a, b). Further statistical analyses confirmed that IL-25 inhibition with LNR125treatment increased expression of type I (IFN-β) and type III (IFN-λ) mRNA (Fig. 2b). We also assessed RV-induced IFN protein expression at 4 d p.i. in $n = 7$ ALI-BECs (moderate to severe asthma) treated with LNR2 (isotype control) or LNR125 IL-25 neutralising antibody. For asthmatic BECs treated with isotype control mAb LNR2, RV did not significantly increase either IFN-β or IFN-λ2/3 protein above baseline (mock-infected cells). In contrast, LNR125 treatment significantly upregulated RV-induced type I/III IFNs compared to mock-infected asthmatic BECs (Fig. 2c). Viral replication/load can affect IFN expression. Viral load was not different between LNR125- and isotype control LNR2-teated cells (Fig. 2d) suggesting that IL-25 directly regulated epithelial cell type I/III IFN expression. We next compared treatment with LNR125 on phosphorylated and total IFN regulatory factor 7 (IRF7), TANK binding kinase 1 (TBK1) protein expression in cell lysates by immunoblot, however, we could not identify differences in protein expression compared to LNR2-

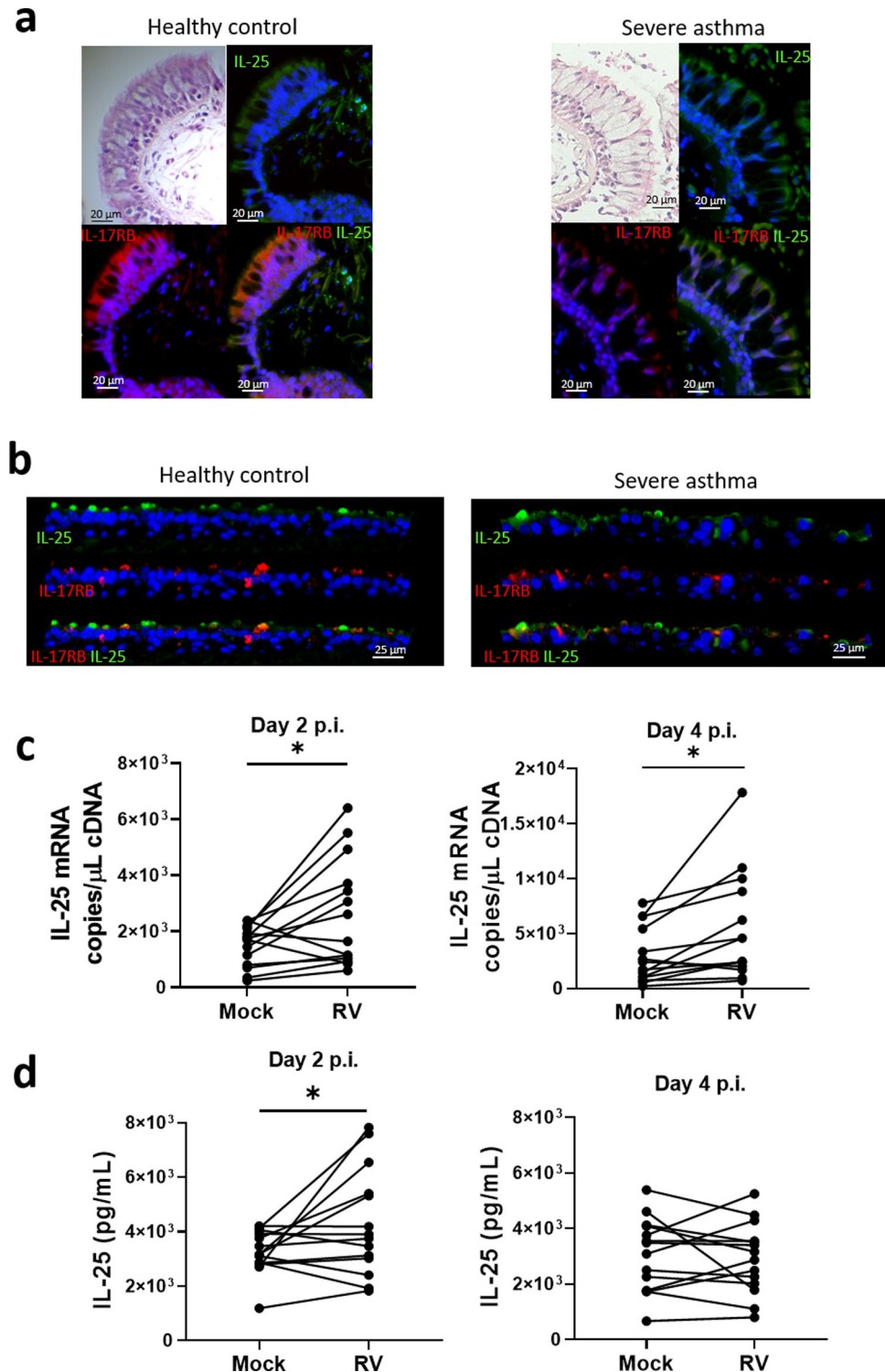

**Fig. 1 Localization and production of IL-25 in bronchial epithelial cells.** H&E and immunofluorescence staining of IL-25 and IL-17RB in (**a**) endoscopic bronchial biopsies or (**b**) air-liquid interface-differentiated bronchial epithelial cells (BECs) from healthy donors and donors with asthma. IL-25 (**c**) mRNA and (**d**) cell-associated protein from RV-A1 infected BECs from donors with asthma at 2- and 4-days postinfection. Representative of $n = 5$ endoscopic bronchial biopsies and BEC cultures, $n = 14$ RV-infected BECs, data analysed by Wilcoxon matched pairs test *$p < 0.05$. Scale bars represent 20 μm and 25 μm, respectively, as indicated.

treated cells (Supplementary Fig. 2c). In summary, blockade of IL-25 during RV-A1 infection upregulated IFN gene and protein expression independently of viral load in ALI-differentiated BECS from donors with asthma.

**IL-25 signalling regulated epithelial innate anti-viral immunity.** We next investigated the effect of LNR125 treatment on IL-25 signalling in $n = 9$ ALI-BECs derived from patients with moderate to severe asthma (Supplementary Table 1). RV

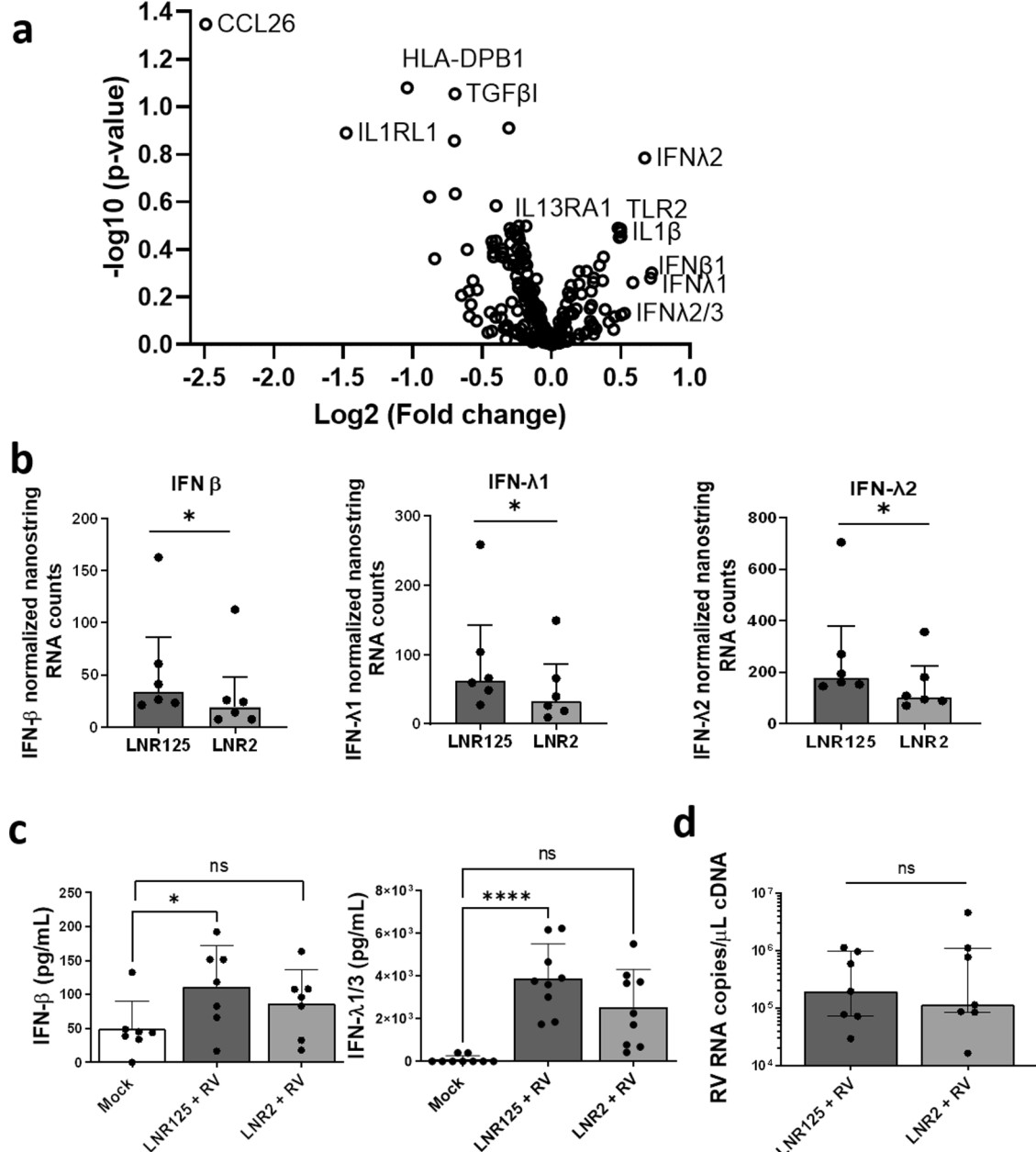

**Fig. 2 IL-25 blockade increased IFN production during RV-A1 infection in differentiated BECs from individuals with asthma.** Differentiated primary bronchial epithelial cells (BECs) from donors with asthma, were treated with either 10 μg/mL of LNR125 or isotype control antibody (LNR2) and infected with RV-A1 and harvested for RNA and supernatants 4 d.p.i. Isolated RNA underwent transcriptomic analysis. (**a**) Volcano plot of differentially expressed genes (DEG) between LNR125 and LNR2 treated RV-infected BECs. (**b**) Differentially expressed mRNA and (**c**) secretion of IFN-β and IFN-λ from asthmatic BECs. (**d**) RV viral load in LNR125 and LNR2 treated RV-infected BECs. $n = 6$ asthmatic BECs for volcano plot and Nanostring RNA quantification. $n = 9$ and, $n = 7$ asthmatic BECs for IFN-λ1/3 and IFN-β ELISA, respectively. (**b**) median $+/-$ IQR analysed by Wilcoxon-signed rank test and (**c**) Friedmen multiple comparisons test, mean with SD, *$P < 0.05$, **$P < 0.01$, ***$P < 0.001$, ****$P < 0.0001$, ns = not significant.

infection in the presence of isotype control LNR2 significantly increased IL-25 mRNA. LNR125 treatment reduced RV-induced IL-25 gene expression such that it was not significantly greater than mock-infected cells identifying IL-25 as another type-2 immune gene reduced by IL-25 blockade in RV-infected BECs. We also measured IL-17RB mRNA expression and noted a near significant ($P = 0.053$) reduction in LNR125-treated cells compared to cells treated with isotype control LNR2 antibody (Fig. 3a). We did not detect a significant reduction in IL-17RB protein in LNR125-treated, RV-infected BECs as assessed by immunoblot with protein loading normalised to β-actin for densitometric quantification (Fig. 3b).

To further investigate the role of IL-25 signalling on BEC innate anti-viral immunity in conditionally reprogrammed-expanded[27] and then differentiated BECs derived from two healthy donors, (5 replicate transwells for each donor combined $n = 10$) were treated with recombinant IL-25. IL-25 treatment significantly increased RV viral load compared to RV infected, untreated cells at 4 d p.i. (Fig. 3c). RV-induced IFN-β and λ gene expression was not significantly different between IL-25- and untreated, RV-infected cells, although we did note a trend for increased IFN-β- and reduced IFN-λ-gene expression in IL-25 treated cells (Fig. 3d). The effect on RV-induced type I/III IFN proteins was consistent - IL-25 treatment reduced expression of

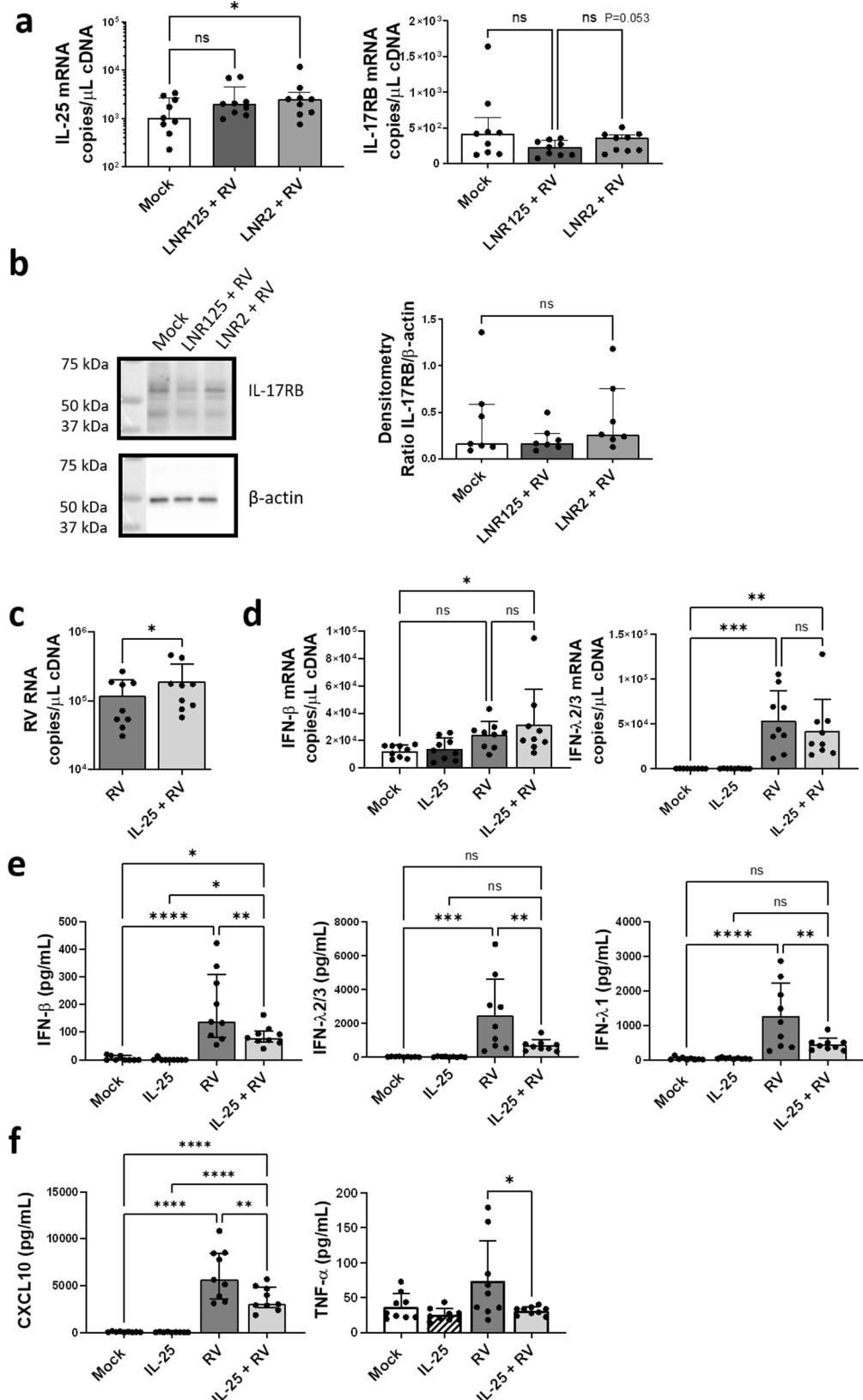

IFN-β, IFN-λ2/3 and IFN-λ1, with suppression of IFN-λ 3-4 fold compared to IFN-β (Fig. 3e). IL-25 treatment also reduced the IFN-induced chemokine CXCL10 and completely ablated the expression of TNF-α protein indicative of broad innate immune suppression (Fig. 3f).

**LNR125 increased the anti-viral response of human coronavirus 229E-infected BECs.** To determine if IL-25 suppresses anti-viral immunity to other (than RV) respiratory viruses BECs from two healthy donors were conditionally reprogrammed to expand before returning to standard ALI conditions for

**Fig. 3 IL-25 negatively regulates antiviral immunity during RV-A1 infection.** Differentiated BECs from nine donors with asthma were treated with 10 µg/mL of LNR125or LNR2, infected with RV-A1 then harvested for total cellular RNA and cellular protein at day 4 p.i. (**a**) IL-25 and IL-17RB gene expression and (**b**) IL-17RB protein expression ($n = 7$ subset of nine asthma donors) measured by immunoblot and quantitation by densitometry using β-actin as loading control. Alternatively, expanded BECs from two healthy donors were differentiated at ALI, treated with 50 ng/mL rhIL-25 and then infected with RV-A1. Total cellular RNA and apical media was collected at day 4 p.i. (**c**) RV viral load was analysed by Taqman qPCR assay. (**d**) Expression of IFN-β and IFNλ2/3 mRNA and (**e**) expression of IFN-β- and IFN-λ-proteins and (**f**) CXCL10 and TNF-α in apical media quantitated using by LEGENDplex. $n = 9$ median $+/-$ IQR, analysed by (**a**, **b**) Wilcoxon matched-pairs t-test (**c**) One-way ANOVA with Holm-Sidak multiple comparisons test mean with SD *$P < 0.05$, **$P < 0.01$, ***$P < 0.001$, ****$P < 0.0001$, ns = not significant.

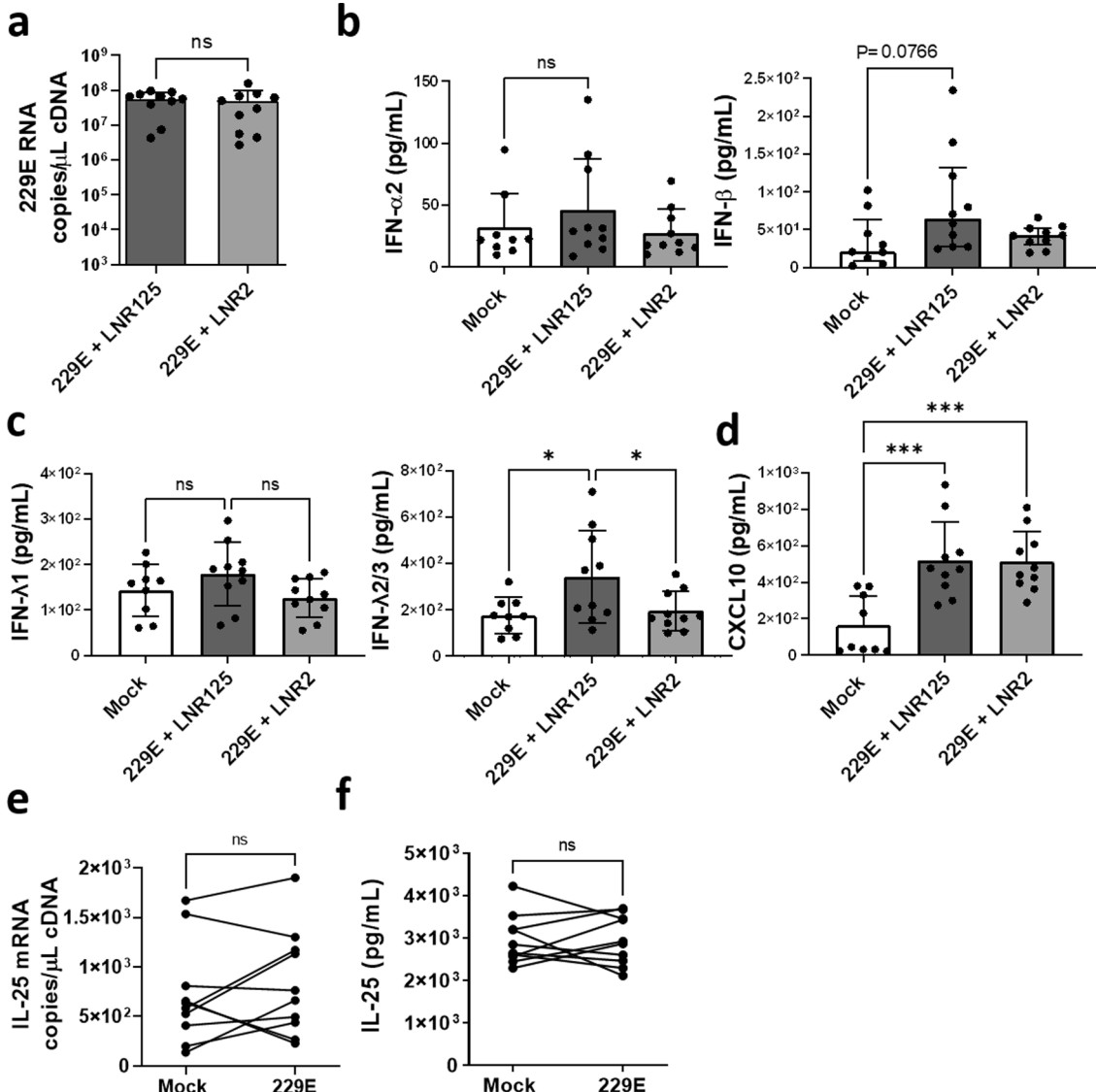

**Fig. 4 IL-25 blockade augments IFN-λ expression during 229E infection in healthy BECs.** Healthy ALI-differentiated CR cells were pre-treated with LNR125 or LNR2 1 day prior to infection with 229E then harvested for total cellular RNA and cellular protein at day 3 p.i. (**a**) 229E viral load was quantified by Taqman qPCR analysis (**b-d**) supernatants were collected in PBS and cytokines were analysed using LEGENDplex. (**e**) IL-25 gene expression and (**f**) protein levels in supernatants quantified by sandwich ELISA. $n = 10$ biological replicates of two healthy CR donors, mean with SD analysed by (**a**) paired T-test, (**b-d**) one-way ANOVA *$P < 0.05$, **$P < 0.01$, ***$P < 0.001$.

differentiation[28]. Five replicate wells each ($n = 10$) were treated with LNR125 then infected with the endemic human coronavirus 229E[29]. We collected cells and apical supernatants at 3 d.p.i. We did not observe any changes in 229E viral load (viral RNA) with LNR125 treatment (Fig. 4a) or difference in type I IFN protein expression (IFN-α2a, IFN-β) although we did note near significant increased ($P = 0.0766$) IFN-β (Fig. 4b). However, LNR125 treatment significantly boosted IFN-λ2/3 protein

expression (Fig. 4c) but did not increase virus-induced CXCL10 (Fig. 4d). No significant difference in IL-25 mRNA or protein levels were detected in this context (Fig. 4e, f).

**LNR125 treatment inhibited RV-induced type-2 cytokine production in vivo.** We have previously reported that RV infection augments aeroallergen-induced lung IL-25 expression associated with increased type 2 lung inflammation and blocking

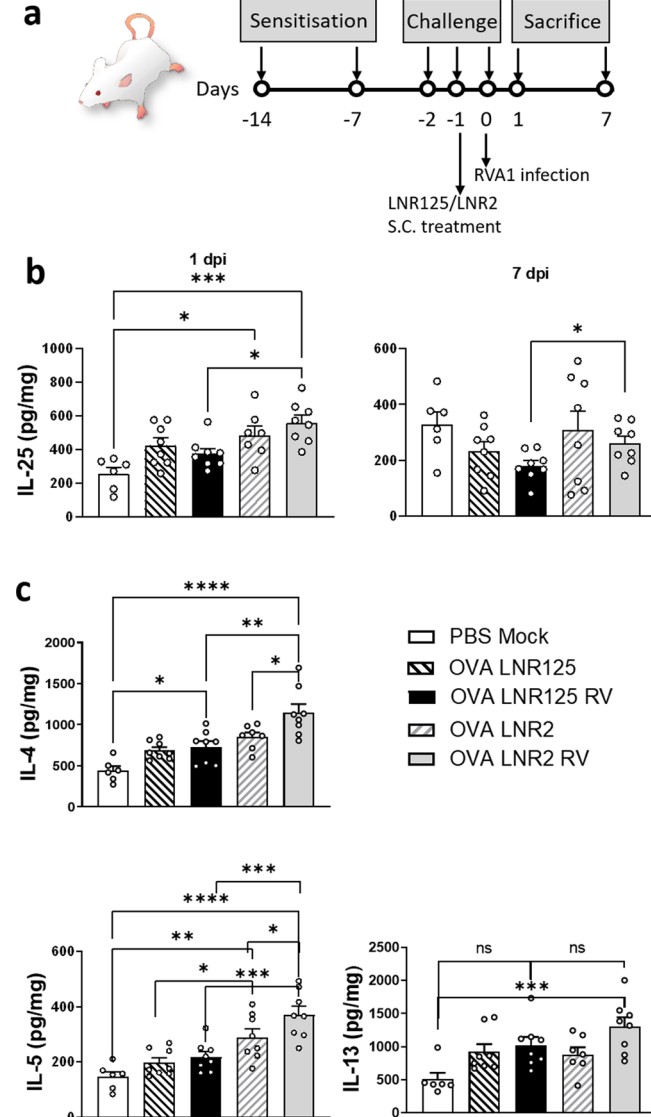

**Fig. 5 IL-25 blockade reduced allergic- RV-A1 induced type-2 cytokine induction.** Mice were sensitised and challenged to ovalbumin (OVA) and on the second day of challenge, treated with LNR125 or LNR2 intraperitoneally followed by infection with RV-A1, 6 h postfinal challenge. (**a**) Schematic of treatment and infection time course. (**b**) Lung IL-25 protein expression at 1- and 7-days postinfection (dpi). (**c**) Lung type 2 cytokine (IL-4, IL-5, and IL-13) protein expression at 1 dpi. Bars represent mean + SEM, $n = 6$–8 mice. *$P < 0.05$, **$P < 0.01$, ***$P < 0.001$, ns = not significant by 1-way ANOVA with Holm-Sidak's correction for multiple analyses.

the IL-25 receptor (IL-17RB) prevented RV exacerbation of allergic airways disease[14]. Utilizing this mouse model, we determined if subcutaneous treatment with LNR125L (or LNR2 isotype control) could similarly reduce type-2 airway inflammation (to directly confirm the role of IL-25 in viral exacerbation of allergic airways inflammation) as well as augment anti-viral immunity. Mice were systemically sensitised and intranasally challenged with OVA to establish allergic airways disease. Mice were administered a single subcutaneous dose of LNR125 or LNR2 isotype control and the following day infected intranasally with RV with samples collected at 1 d p.i. and 7 d p.i. (Fig. 5a). In isotype LNR2-treated mice, OVA challenge increased IL-25 lung protein compared to negative controls (PBS mock) and this was

increased further by RV infection as predicted. LNR125 treatment suppressed both allergen (OVA LNR125) and allergen + virus (OVA LNR1255 RV) induced IL-25 lung protein. LNR125 suppression of allergen and virus-induced IL-25 was also evident at 7 d.p.i. (Fig. 5b). We next investigated downstream inflammatory mediators of IL-25 which promote allergic inflammation. RV infection augmented OVA-induced type 2 cytokine (IL-4, IL-5, and IL-13) production in LNR2 treated mice. LNR125 treatment reduced OVA + RV induced IL-4 and IL-5 protein in BAL, with a trend-non significant reduction in IL-13 compared to LNR2-treated controls such that only OVA LNR2 RV mice had significantly increased BAL IL-13 compared non-allergic, uninfected (PBS Mock) mice (Fig. 5c).

**LNR125 treatment enhanced anti-viral immunity during viral exacerbation of allergic airways disease.** After confirming that LNR125 reduced type-2 cytokines related to RV-exacerbated allergic airways disease, we next evaluated anti-viral IFN proteins in BAL. LNR125 enhanced the production of IFN-β in the RV-exacerbated OVA model at 1 d.p.i compared to the mock-infected and RV-infected LNR2 controls. There was no difference in the level of RV infection induced IFN-λ protein expression in LNR125 and LNR2-treated mice BAL – both RV-infected groups had highly significant induction of IFN-λ2/3 (Fig. 6a). Treatment with LNR125 reduced lung viral load compared to LNR2-treated mice at 1 d.p.i. confirming that LNR125-mediated innate immune re-calibration enhanced anti-viral immunity in vivo (Fig. 6b).

**IL-25 blockade reduced type-2 airway inflammation.** Having examined cytokine and interferon responses we next investigated the role of IL-25 in regulating infiltration of immune cells to determine if the reduced inflammatory cytokines from the BAL corresponded to reduced airway infiltration of cells collected by BAL[26]. RV-infected mice treated with LNR125 had reduced airway infiltration of total cells at 1 d.p.i. compared to isotype control antibody LNR2 treated mice (Fig. 7a). In comparison, RV-infected mice treated with OVA-LNR125 had a trend in reduced total BAL, lymphocytes, and eosinophils at 7 d.p.i ($p = 0.08$, $p = 0.07$, and $p = 0.16$, respectively) compared to RV-infected OVA-LNR2 treated mice (Fig. 7a–c). There was no reduction in macrophages at 1- or 7-d.p.i., however macrophages displayed a trend to be reduced in OVA LNR2 RV compared to OVA LNR125 RV (Fig. 7d). Airway neutrophilic inflammation is caused by viral infection and this was reduced in LNR125treated mice such that only OVA LNR2 RV mice had increased BAL neutrophils compared to untreated/infected (PBS mock) mice at day 7 p.i. (Fig. 7e).

**Discussion**
Epithelial cell-expressed IL-25 promotes allergic diseases such as asthma and is increased during viral asthma exacerbations[14,30,31]. IL-25 stimulates type 2 inflammation and contributes to airway obstruction by triggering bronchoconstriction, mucus production, and infiltration of inflammatory cells into airways. The need for efficient control of infection by airway epithelial cells to limit the capacity of viruses to provoke airway inflammation in the context of asthma exacerbations has led to the discovery of epithelial cell-intrinsic delayed and deficient anti-viral immunity[23–25]. IL-25 is constitutively expressed by airway epithelial cells[19] making it a candidate for regulating epithelial cell innate immunity which led us to investigate the role of IL-25 on antiviral immunity in asthma. First, we confirmed that IL-25 is constitutively expressed in vivo in human airways (endobronchial biopsies), and this was replicated in differentiated primary human BECs. We noted abundant constitutive expression of IL-25

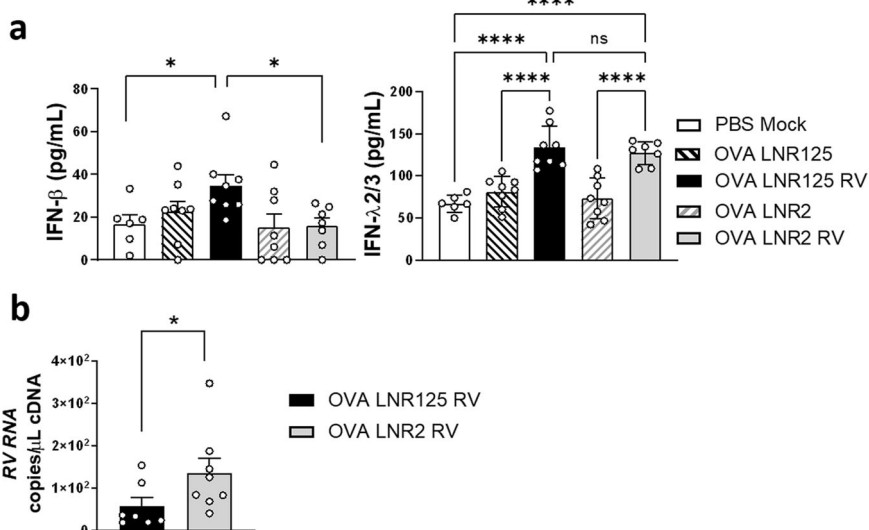

**Fig. 6 IL-25 blockade enhanced IFN-secretion and reduced viral load in allergic mice and mice with RV-induced exacerbations.** Mice were sensitised and challenged to ovalbumin (OVA). On the same day as the second challenge dose, mice were treated with LNR125 or LNR2, intraperitoneally followed by infection with RV-A1, 6 h postfinal challenge. (**a**) Lung tissue was collected 1 day postinfection for IFN analysis by ELISA. (**b**) Viral load was quantified by Taqman qPCR analysis of apical lung lysates. Bars represent mean + SEM, $n = 6$–8 mice. *$P < 0.05$, **$P < 0.01$, ***$P < 0.001$, ****$P < 0.0001$, ns = not significant by (**a**) or 1-way ANOVA with Holm-Sidak's correction for multiple analyses (**b**) or Mann Whitney test for non-parametric analysis.

protein at the apical surface airway epithelial cells from both healthy donors and subjects with asthma. A similar expression pattern was noted for the IL-25 receptor with evidence of colocalization of IL-25 and IL-17RB, supporting a role for IL-25 autocrine/paracrine signalling in directly regulating epithelial cell-mediated mucosal immunity, in addition to indirectly via activation of IL-17RB-expressing mucosal-resident innate immune cells such as type 2 innate lymphoid cells[32] and plasmacytoid dendritic cells[33]. The endobronchial biopsies we selected to examine predominantly contained intact airway epithelium and IL-17RB-expressing immune cells were not observed. Future studies could examine biopsies with lung tissue and include CD45 co-staining to identify leucocytes. Despite the abundant constitutive expression, we were able to detect increased IL-25 mRNA and protein (cell associated) expression by RV infection of ALI-BECs derived from asthmatic donors. We could not reliably detect secreted IL-25 in apical supernatants consistent with substantial binding of IL-25 to the apical surface of epithelial cells. A recent study in nasal epithelial cells has reported that IL-25 is induced during influenza A virus infection, and that pretreatment with IFN-α reduced IL-25 gene expression suggesting that type I IFN antagonised virus-induced epithelial cell IL-25[34].

To determine if the reciprocal is true – IL-25 negatively regulates anti-viral responses - we used a potent anti-IL-25 mAb (LNR125) to determine if blocking IL-25 produced by BECs from patients with moderate to severe asthma augmented IFN production during RV infection. To gain an overview of the effect on BEC innate immunity we used Nanostring Immune transcriptomic analyses and indicated that blocking IL-25 promoted expression of innate anti-viral interferons (type I/III IFNs) and ISG expression whilst suppressing expression of immune pathways that antagonise anti-viral immunity – type 2 immunity[35] and TGF-β[36]. Multiple mechanisms by which type 2 immune pathways interfere with innate anti-viral responses in the context of asthma have been described[37,38]. This is the first study to provide evidence that a therapeutic anti-IL-25 mAb will directly boost airway epithelial cell anti-viral immunity and points to IL-25 blockade as being particularly beneficial for viral asthma exacerbations where a therapy with dual activity – block type 2

inflammation and promote anti-viral immunity – would be desirable.

The precise mechanism by which IL-25 inhibits anti-viral innate immunity remains to be elucidated. Nanostring mRNA expression analysis suggested up-regulation of key molecules in anti-viral signalling pathway including TBK-1, IRAK2 and IRF7. However, we were unable to confirm this at the protein level due to the heterogeneity of donor BEC responses. IL-25 signalling can activate multiple transcription factors including p38 mitogen-activated kinase, c-Jun, NF-κB and STAT5[39,40]. Of these, STAT5 is a candidate for interfering with STAT1 mediated interferon responses[41].

Biologics (mAbs) specifically targeting type 2 immune pathways have provided clinical benefit in reducing the frequency of asthma exacerbations and there are now several approved for use in asthma including omalizumab (anti-IgE mAb)[42,43], dupilumab (anti-IL-4Rα)[6,44], mepolizumab and reslizumab (anti-IL-5)[10,45–48] and benralizumab (anti-IL-5R)[49,50]. The use of these biologics is limited to asthmatics who cannot control symptoms with corticosteroids and satisfy immunological criteria: omalizumab is approved for use in atopic asthma (high IgE). For the type 2 cytokine-targeting mAbs there must be elevated blood eosinophils. Based on these criteria many patients with severe, difficult to treat asthma are not eligible for treatment with these mAbs highlighting the need for further research into understanding the immunological aetiology of different disease phenotypes/endotypes in severe asthma[51]. Since the majority of severe asthma exacerbations are triggered by a respiratory virus infection[3], targeting cytokines that emanate from infected airway epithelium has become a focus. Three such cytokines – IL-25, IL-33 and TSLP – have been of particular interest given their role in stimulating airway inflammation in models of severe asthma and asthma exacerbations[14,52,53] with increasingly detailed dissection of their functions revealing distinct effects on specific immune cell populations[54]. Both anti-TLSP (tezepelumab) and anti-IL-33 (itepekimab) have reported phase 2 clinical data in asthma: anti-TLSP showing the reduction in the frequency of asthma exacerbations[13] and anti-IL-33 provided some protection against loss of asthma control[55].

Less progress has been made with treatments that stimulate anti-viral immunity. Inhaled IFN-β has previously been tested in

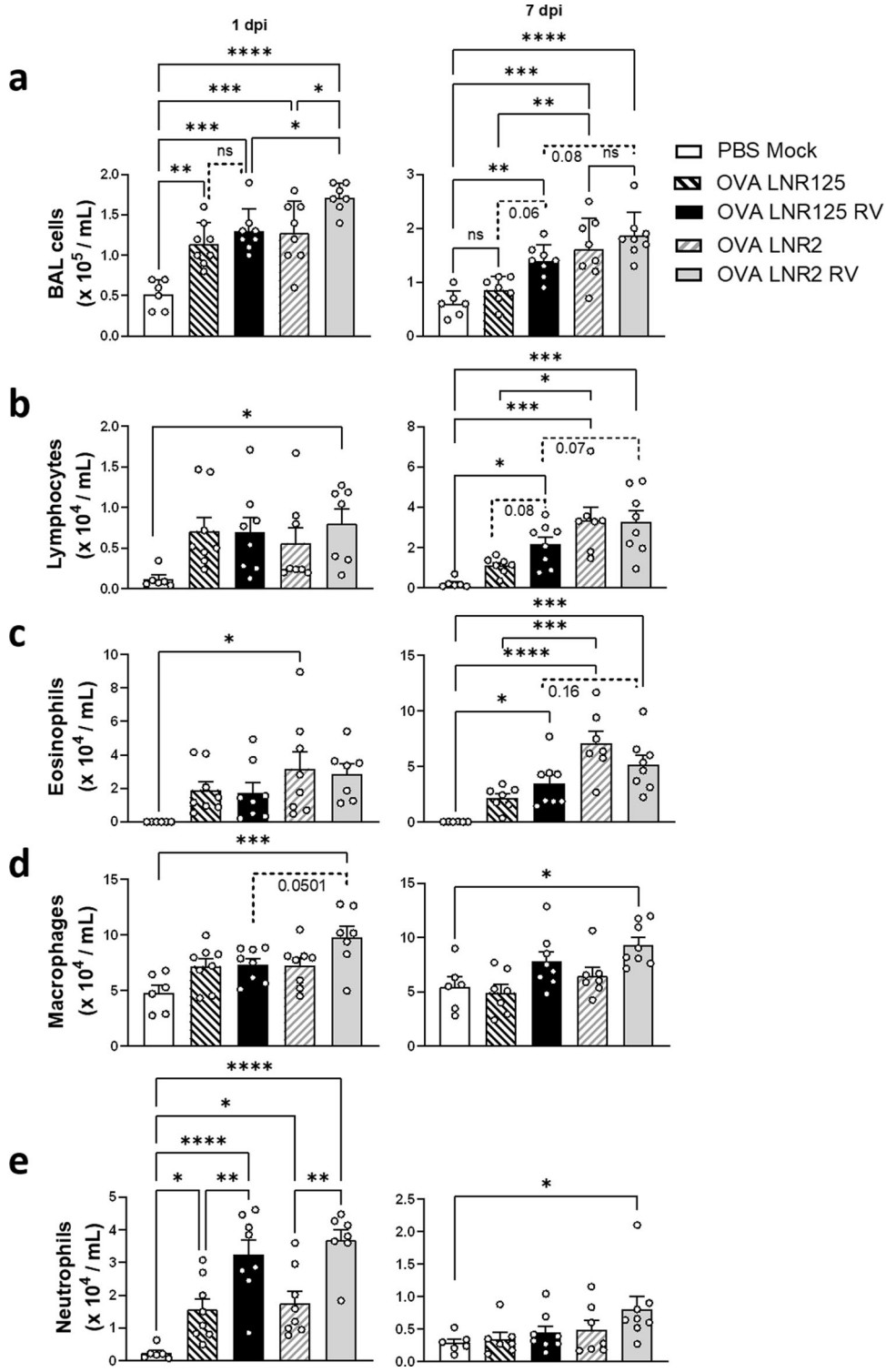

**Fig. 7 IL-25 blockade reduced immune cell infiltration at peak- and resolution of inflammation in allergic mice and mice with RV-induced exacerbation.**
Mice were sensitised and challenged to ovalbumin (OVA). On the same day as second challenge dose, mice were treated with LNR125 or LNR2, intraperitoneally followed by infection with RV-A1, 6 h postfinal challenge. Bronchoalveolar lavage (BAL) was performed to innumerate immune cell counts at day 1- and 7- postinfection. (**a**) Total BAL counts, (**b**) lymphocyte, (**c**) eosinophil, (**d**) macrophage and (**e**) neutrophil cell counts. Bars represent mean + SEM, $n = 6$–8 mice. *$P < 0.05$, **$P < 0.01$, ***$P < 0.001$, ****$P < 0.0001$ by mixed effects model of 2-way ANOVA with Holm-Sidak's correction for multiple analyses. $P$ values > 0.05 denoted by exact $P$ values and broken lines.

a clinical trial for individuals with asthma during respiratory viral infections. Assessment of the clinical benefit of inhaled IFN-β was confounded by the lack of virus-induced exacerbations detected in this study and therefore a reduction of asthma symptoms was not detected. Analysis of a subgroup with the severe disease did show IFN-β treatment reduced symptom severity[56]. One potential reason for the limited efficacy of inhaled IFN-β is limited duration of in vivo bioactivity of the recombinant protein. Blocking IL-25 is evidence to support innate immune-targeting approaches that re-calibrate mucosal innate immunity to improve the capacity for endogenous IFN production during viral infection. Other innate immune-stimulating approaches support this strategy. We have reported that TLR2-agonist innate immune priming also re-calibrates BEC response to RV infection, characterised by rapid NF-κB activation and IFN-λ production that enhances control of viral infection in vitro and in vivo[28].

We showed that LNR125-IL-25 blockade increased IFN-β and IFN-λ mRNA and protein expressed by RV infected BECs from subjects with moderate to severe asthma, a disease phenotype associated with deficient virus-induced epithelial cell IFN production[25]. The focus of this study was to understand the effect of IL-25 on anti-viral immunity in asthma to determine the therapeutic potential of IL-25 blockade for viral asthma exacerbations. Therefore, we did not compare IL-25 blockade and anti-viral responses of BECs from asthmatic subjects with BECs from healthy donors. We did note that the RV-infected asthmatic BECs used in this study, treated with isotype control antibody, failed to significantly increase expression of type I/III IFNs by 4-day postinfection which is consistent with a deficient response, which could be improved by mAb-mediated inhibition of IL-25. Immune transcriptome analysis of LNR125 treated BECs from individuals with asthma indicated that blocking IL-25 restored antiviral immunity with evidence of upregulation of ISGs, such as IRF7 and TBK1 and interleukin-1 receptor-associated kinase 2 (IRAK2). Protein validation of total and phosphorylated TBK1 and IRF7 by immunoblot could not detect a difference in protein expression and therefore did not provide confirmatory evidence for the transcriptomic data. We attributed this to the low sensitivity of western blot for quantifying transcription factors, particularly in ALI-differentiated BEC-RV infection models in which only a small number of cells are infected[25]. Indeed, patchy patterns of viral infection has been reported in airway epithelial cells during respiratory viral infection[57]. Infected cells are eliminated by a process of extrusion (ejected from the epithelium) or epithelial cell phagocytosis (efferocytosis)[58]. We examined LNR125 treatment in healthy BECS infected with an endemic coronavirus strain 229E. We observed enhanced production of IFN-λ2/3. Further, we found a trend in upregulation of IFN-λ1, IFN-β, IFN-α2, and IFN-γ. We can conclude that treatment with LNR125 induced an earlier innate immune response in healthy BECs infected with 229E, implicating a potential role for IL-25 blockade in boosting anti-viral immunity during coronavirus infection.

In addition to IFNs, we found a trend toward increased IL-1β, a component of the inflammasome which can suppress IL-25 cytokine production during helminth infection[59]. Inversely, IL-25 has previously been reported to be a negative regulator of proinflammatory cytokines secretion of IL-1β, TNF-α, and IL-6 In monocytes[60]. This data implicates IL-25 suppression of anti-viral immunity extends to multiple respiratory viruses including coronavirus, RV, and influenza[34]. We also examined the induction of IL-25 and effects on antiviral immunity during CoV 229E infection. Unlike RV, we did not observe 229E-increased IL-25 expression by differentiated primary human BECs, although blocking IL-25 did increase virus-induced IFN-λ. This observation is consistent with RV having greater capacity to promote type 2 inflammation and might underlie the greater prevalence of

this virus in triggering asthma exacerbations. However, the RV-induced IL-25 and 229E induced IL-25 datasets are not directly comparable: the data for RV-induced IL-25 was derived from BECs from 14 donors and assessed at two-time points (2 and 4 days postinfection). Although statistically significant RV-induced IL-25 expression, this was heterogeneous with some donors' BECs not exhibiting increased IL-25 with RV infection. For 229E, we used BECs from two donors (replicates for each donor) and examined induction at the single time point (3 days postinfection) and did not observe viral increased IL-25 expression. It is possible that if we examined cells from more donors and additional timepoints for 229E infection we would detect increased expression of IL-25 indicating that 229E, like RV, can stimulate type 2 inflammation and cause asthma exacerbation. Further studies that directly compare RV and 229E-induced IL-25 in BECs from the same donors are required to formally address this.

Previous studies have shown that IL-25 or IL-17RB blockade during respiratory viral infection in vivo has reduced immune cell infiltration and pro-inflammatory mediators in the BAL[14,30,61]. Two IL-17 family cytokines signal via IL-17RB – IL-17E (IL-25) and IL-17B. Therefore blocking IL-17RB was not definitive proof that IL-25 was the cytokine driving lung type 2 inflammation, although it was highly likely given lung expression and established role in type 2 immunity for IL-25[14] versus low lung expression and distinct biological activity for IL-17B[62]. Using a mouse model of RV-exacerbation of allergic airway disease, we found LNR125 reduced total BAL cell counts, and secretion of IL-4, IL-5, and IL-25. This confirmed a central role for IL-25 in regulating virus induce airway inflammation. In contrast to in vitro epithelial cells directly treated with anti-IL-IL-25, in vivo we found systemic administration of LNR125 upregulated lung IFN-β, but not IFN-λ. This is consistent with the specific role of type III IFN-λs in epithelial cell anti-viral immunity whilst systemic inhibition of IL-25 in vivo promoted expression of IFN-β by multiple cell types[63].

IFN-γ has been previously reported to be suppressed by IL-25 during helminth infection, however, we are the first to show IL-25 regulates epithelial-derived virus-induced IFNs during allergic inflammation[64,65].

IL-25 induces and amplifies type-2 inflammation through activation and recruitment of lymphocytes and granulocytes[66,67]. Moreover, a feed-forward mechanism has been identified within human lung tissue where IL-4 upregulated IL-25 and IL-17RB expression[68]. We found LNR125 treatment reduced IL-17RB mRNA expression within our in vitro model, indicating that LNR125 treatment could disrupt this feed-forward propagation of type-2 inflammation. In our mouse model, we observed LNR125 reduced total BAL, lymphocytes, and eosinophil recruitment 7 d.p.i., supporting the role of IL-25 in amplifying type-2 inflammation. During an RV exacerbation of allergic airways disease, LNR125 treatment inhibited the expression of IL-4, IL-5, and IL-25, which is entirely consistent with a previous study in this model using mAb blockade of the IL-25 receptor (IL-17RB)[14].

Increased inflammation in asthma has been associated with airway remodelling and reduced BEC differentiation[69]. One insight from the IL-25 blockade transcriptomic data in BECs from donors with asthma highlighted that IL-25 regulates TGF-β signalling; a contributor to airway remodelling[70]. While IL-25 blockade did not regulate any isoforms of TGF-β, we found LNR125 treatment reduced the inducer of TGF-β (TGF-βI), while upregulating the negative regulator SKI. This implicates that IL-25 can indirectly regulate TGF-β expression. A study conducted in primary BECs treated with an anti-TGF-β antibody reported increased IFN-β and IFN-λ1 production when stimulated with poly IC (a viral mimic)[71]. Therefore, the enhanced IFN secretion

we report with LNR125 may be through multiple mechanisms such as suppression of TGF-β signalling, in addition to the enhanced ISG expression. Future studies will investigate the interplay of IL-25 and TGF-β in BECs during respiratory viral infection.

In conclusion, we show that IL-25 blockade improved antiviral immunity during respiratory viral infection. In BECs from donors with asthma, this was through inhibiting the suppressive effects of IL-25 on IFN production and ISG expression and reducing type-2 inflammation. Our in vivo model of viral exacerbation of allergic airways disease further showed reductions in type 2 cytokines and IL-25 in the lung associated with increased IFN-β production and reduced lung viral load. Finally, we showed that IL-25 blockade during coronavirus infection upregulated IFN-λ2/3. Therefore, IL-25-induced airway inflammation combined with suppression of epithelial cell anti-viral immunity identify IL-25 as central mediator of viral asthma exacerbations.

## Methods

**Ethics statements.** Primary BEC brushings were provided by Professor Peter A. B. Wark (The University of Newcastle) in compliance with the University of Newcastle and Hunter New England Area Health ethics committee (05/08/10/3.09) with written, informed consent prior to collection. Animal experiments were conducted in accordance with the NSW, Australia Animal Research legislation on protocol A-2016-605, reviewed and approved by the University of Newcastle Animal Care and Ethics Committee.

**Air-liquid interface culture of BECs.** BECs were obtained from moderate-severe persistent asthmatic donors or donors with GOLD stage 2-3 COPD as defined by asthma and COPD guidelines, respectively. Primary BECs were cultured until confluent then differentiated at air-liquid interface (ALI), as previously described[25,72]. Alternatively, BECs were obtained from healthy donors and conditionally reprogrammed (CR) with rho-associated protein kinase (ROCK) inhibitor (final concentration 10 μM) in combination with irradiated 3T3 feeder cells in monolayer cultures[73,74]. CR media consisted of 1:2 ratio of DMEM (high glucose + L-glutamine)/Ham's F12 supplemented with 5% FCS, hydrocortisone (400 ng/mL), insulin (5 μg/mL), rhEGF (10 ng/mL), cholera toxin (8.4 ng/mL), adenine (23.9 μg/mL), and 0.2% penicillin streptomycin[27]. Expanded BECs were weaned off the ROCK inhibitor and seeded onto polyester transwell inserts and differentiated. All differentiated donor demographics are described in Supplementary Table 1. A selection of listed donors was used for IL-25 blockade experiments.

**Antibody treatment with RV-A1 or 229E infection.** Anti-IL-25 antibody LNR125 and LNR126 (developed by Abeome Corporation, now Lanier Biotherapeutics, Athens, GA, USA) was generated using the AbeoMouse™ platform which allows the selection of affinity matured B cells via cell surface antibody selection (Supplementary Fig. 3a). Neutralising potency of anti-IL-25 antibodies against mammalian cell expressed human- and mouse IL-25 was determined by calculating $IC_{50}$ in HT29 cell reporter assay and indicated that LNR125 neutralised both human and mouse whereas LNR126 neutralised human IL-25 only. A sandwich ELISA format was used to perform epitope binning assays and determine that LNR125 does not interfere with LNR126 IL-25 binding and vice versa and therefore this antibody pair identified as suitable for human IL-25 ELISA development (Supplementary Fig. 3b).

Day one prior to infection, the ALI-BECs was treated basally with the anti-IL-25 monoclonal antibody LNR125, or matched IgG isotype control LNR2 (formerly Abeome Corporation now Lanier Biotherapeutics, USA) at 10 μg/mL in BEBM minimal media (BEBM + 1% ITS and 0.5% linoleic acid-BSA) (Lonza, Switzerland). BEC cultures were infected apically with RV-A1 (MOI 0.1) for 2 h at 35 °C. Following infection, the virus inoculum was removed, and the apical surface was washed twice with PBS. Minimal media containing LNR125, LNR2, or media alone was placed apically and refreshed basally, and cells were incubated at 35 °C until day 2 or 4 postinfection (d.p.i.). Apical supernatants and basal media were collected at indicated time points and stored at −80 °C for downstream protein analysis. Half transwells were lysed in RLT buffer (Qiagen, Germany) containing 1% 2-mercaptoethanol or RIPA buffer containing protease inhibitor cocktail (Roche, Switzerland). Lysate was stored at −80 °C.

Separately, 10 μg/mL of LNR125, LNR2, or ALI-BEC media used throughout differentiation was applied apically and basally 1 day prior to infection. Apical media was removed and BECs were infected with 229E (MOI 0.1) 2 h at 35 °C. Following infection, the virus inoculum was removed, and the apical surface was washed twice with PBS. Basal media was replaced and supplemented as with RV infection. Apically secreted mediators were collected in PBS on day three and lysates collected as above.

**Mouse model of viral exacerbation of allergic airways disease.** 6–8-week-old, wild-type, female BALB/c mice were obtained from Australian Bioresources (ABR, Moss Vale, NSW), sensitised with 50 μg chicken Ovalbumin (OVA) protein in 1% alhydrogel intraperitoneally (i.p.) on day −14 and day −7 followed by intranasal challenge (i.n.) with 50 μg of low LPS OVA in 30 μL of PBS (controls receive PBS alone) on 3 consecutive days (Days −2, −1, and 0) to induce allergic airway inflammation. 250 μg LNR125 or LNR2 (in 100 μL) was administered subcutaneously (s.c.) on day −1. Mice were then infected i.n. with $2.5 \times 10^6$ TCID$_{50}$/mL of RV-A1 or mock infected with PBS, 6 h after final OVA challenge. On day 1 and day 7 post-infection, bronchoalveolar lavage samples (BAL fluid and leukocytes) were collected, apical lung lobe tissue was collected for RNA extraction, and the remaining lung tissue was snap-frozen for cytokine analyses.

**RNA extraction and gene expression analysis for quantitative real-time (qRT)-PCR.** RNA was extracted using the miRNeasy kit (Qiagen, Germany) following the supplier's protocol. RNA concentrations were determined by Nanodrop and 200 ng (cells) or 500 ng (tissue) was reverse transcribed using the high-capacity cDNA reverse transcriptase kit (ThermoFisher Scientific, USA). Quantitative PCR was performed on a QuantStudio 6 machine (Applied Biosystems, United States) using Taqman mastermix (Applied Biosystems, United States) and customized TaqMan FAM-TAMRA primers and probes purchased from IDT (Supplementary Table 2). Gene copy numbers were normalized to housekeeping gene 18 S and quantitated using a standard curve.

**Immune transcriptome analysis.** Extracted RNA was hybridized to the human immunology v2 GX code set (Nanostring, United States) as per manufacturer's instructions with 50 ng of RNA. Raw data were imported into the Nanostring nCounter advanced analysis software v2.0 for normalisation based on positive and negative controls, and GENorm selection of housekeeper genes. Normalised data were then exported and graphed as digital mRNA counts for exact copy number or were expressed as log2 fold change ratio against -log10 Bejamini-Yekutieli-corrected p-values for volcano plot data visualisation[28].

**Cytokine and chemokine protein analysis.** BEC ALI apical supernatants were assessed for IFN-λ1/3, (R&D, United States) and IFN-β (PBL assay sci, United States) protein expression by ELISA, as per the manufacturer's instructions.

Human IL-25 was measured in BEC apical supernatants and cell lysates by sandwich ELISA (Formerly Abeome Corporation, now Lanier Biotherapeutics, United States).

Cell debris from protein lysates was removed by centrifugation at 9000×g for 10 min at 4 °C and protein concentration was determined by BCA assay (ThermoFisher, USA).

96 well plates were coated with LNR126 capture antibody in PBS and stored at 4 °C overnight. Plates were washed 3X in PBS and 0.075% Tween-20 and blocked with reagent diluent (R&D systems, USA) at room temperature for one hour. The standard was prepared with recombinant IL-25 (R&D Systems) with a range of 250–1.95 pg/mL in reagent diluent. Samples were diluted in a 1:4 combination of RIPA buffer and reagent diluent. Biotinylated LNR125 detection antibody (LNR125-HRP) was diluted in blocking buffer and incubated for 1 h at room temperature. For detection, 1-step Ultra TMB reagent (Thermofisher, USA) was added. Readings were taken at 5, 10, 15, 20, and 30 min at 633 nm during development with gentle agitation before each reading. Peak absorbance readings were taken and normalized to BCA protein concentration.

Mouse IL-4, IL-5, IL-13, IL-25, IFN-β, and IFN-λ2/3 were quantified by Duoset ELISA (R&D Systems, United States).

Cytokine and chemokine quantification of conditionally reprogrammed cells for IFN-β, IFN-λ2/3, IFN-λ1, IFN-γ, IFN-α2, CXCL10, IL-1β, and IL-6 was measured using the LEGENDplex human anti-virus response multiplex flow cytometry panel (Biolegend, United States) as per the manufacturer's instructions. Data were acquired with the FACS Canto II (Beckman Coulter; United States) and analysed with LEGENDplex v8.0 software (Biolegend, United States).

**Immunofluorescence.** We have previously confirmed the differentiation status (stratified, active cilia, mucin-positive goblet cells) of the ALI-BEC cultures from which sections were subsequently obtained for immunofluorescence-based analyses of IL-25 and IL-17RB expression for this study[25]. Paraffin-embedded endoscopic lung biopsies and ALI-BEC sections were deparaffinised in xylene, then rehydrated in ethanol before being subjected to antigen retrieval in sodium citrate buffer. Slides were washed in TBS-T then were blocked for 1 h in 5% donkey serum/5% casein solution in TBS-T in a humidified chamber. The blocked sections were incubated overnight at 4 °C with anti-IL-17RB (MAB1207, R&D Systems) and anti-IL-25 (BAF1258, R&D Systems, USA) in 2% donkey serum/2% casein in TBS-T. Primary antibodies were washed off and HRP-488 (405235, Biolegend, USA) and Alexa-fluor-594 (SAB4600407, Sigma Aldrich, USA) were applied in the dark for 2 h at room temperature. Secondary antibodies were washed off and mountant containing DAPI was applied prior to imaging with the Axio imager M2 fluorescent microscope and analysing with Zen 2 software (Carl Zeiss AG, Germany).

**Western blot**. Cell debris from protein lysates was removed by centrifugation at 9000×g for 10 min at 4 °C and protein concentration was determined by BCA assay (ThermoFisher, USA). Five micrograms of protein was resolved by 4–15% TGX Stain-free SDS PAGE gel (Biorad, USA) and semi-dry transferred onto nitro-cellulose (Biorad, USA). Membranes were blocked in 3% BSA 2% skim milk in TBS-T for 1 h at room temperature, then incubated overnight at 4 °C with anti-IL-17RB or β-actin (ab8227, Abcam). After extensive washing, membranes were incubated with HRP-conjugated secondary antibodies for 1 h. Membranes were developed with Supersignal west femto (ThermoFisher, USA) and imaged on a Chemi-Doc (Biorad, USA).

**Statistical analyses and reproducibility**. Results and error bars were presented as +/− standard deviation (s.d.) or standard error of the mean (s.e.m.) or median +/− interquartile range (IQR), as indicated. In all experiments, $n$ represents the number of individual BECs donors or mice. Wilcoxon matched-pairs t-tests were applied to determine statistical significance for differences between two groups. Parametric in vitro data was analysed using Friedman multiple comparisons test or one-way ANOVA where applicable Holm-Sidak's correction for multiple analysis. Parametric mouse data were analysed using one-way or two-way ANOVA where appropriate with Holm-Sidak's correction for multiple analyses. Non-parametric mouse data were analysed by a Mann-Whitney test. Statistical significance was set at $*p < 0.05$, $**p < 0.01$, $***p < 0.001$, $****p < 0.0001$. Analyses were performed using GraphPad Prism v8.3 software (GraphPad, United States).

**Reporting summary**. Further information on research design is available in the Nature Research Reporting Summary linked to this article.

## Data availability

All data generated or analysed during this study are included in this published article (and its supplementary information files). Source data underlying the graphs in the paper can be found in the Supplementary Data file.

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

## Author contributions

T.C.W., P.A.B.W., C.L.G., D.A.K., T.V., C.L.J., K.A., R.A.S., J.L.G., and N.W.B. contributed to the conception and design of the work. K.S.N., A.T.R., P.C.V., C.E., C.L.J., D.A.K., and P.A.B.W. provided technical assistance on cell culture and scientific input on experiments. T.C.W., S.L., C.E., J.L.G., and N.W.B. acquired, analysed and interpreted results. K.A., T.V., R.A.S., J.L.G., and N.W.B. reviewed work for intellectual content. T.C.W., S.L., J.L.G., and N.W.B. co-wrote and reviewed the manuscript. T.C.W., J.L.G., and N.W.B. agree to be accountable for all aspects of the work in ensuring that the questions related to the accuracy or integrity of any part of the work are appropriately investigated and resolved.

## Competing interests

NWB reports grants, personal fees for consultancy and other (stock options) from Abeome Corporation/Lanier Biotherapeutics during the conduct of the study; has a patent PCT WO2017160587A1. KA is an employee of Abeome Corporation and owns stock/options in Abeome Corporation and Lanier Biotherapeutics. RAS reports personal fees for consultancy and owns stock/options in Abeome Corporation and Lanier Biotherapeutics; has a patent PCT WO2017160587A1. CLJ is an employee of Lanier Biotherapeutics and owns stock/options in Abeome Corporation and Lanier Biotherapeutics; has a patent PCT WO2017160587A1. The remaining authors declare no competing interests.
