## [Peer Review File · Communications Biology]

Reviewers' comments:

Reviewer #1 (Remarks to the Author):

This is a well performed study which reports the ability of IL-25 to decrease anti-viral immune responses to RV and as such the ability of an anti-IL-25 antibody to promote viral clearance including through enhanced IFN production and ISG expression and reduced T2 inflammation. Both ex vivo and in an in vivo murine model IL-25 reduced anti-viral immunity and increased viral load and conversely anti-IL-25 restored anti-viral immunity and reduced viral loads. Given the expected limitations of performing research in (outbred) humans, there are only a few minor considerations.

1. IL-17RB expression (figure 1A). It might be expected in the biopsy specimens to see this being expressed on mast cells, ILC2s, eos, and others. No obvious "red" stained cells are observed in this figure.
2. There is a problem with the IL-25 in house EIA insofar as it is using the same antibody for capture and enzyme-linkage, the problem being that the former antibody is likely to stoichiometrically block available epitopes for the latter antibody. This may be contributing to the poor data displayed in 1D (where all the day 4 samples and half the day 2 samples show no increase).
3. Figure 2A is impressive. Figure 2 is missing a label for "D".
4. The authors should discuss more reasons as to why the response to RV and CoV are so different. Perhaps this points to the more T2-inducing effects of RV that are being abrogated by the anti-IL-25 antibody. Did the CoV have a similar impact on the T2 cytokines (possibly with the high T2 response to RV triggering an anti-viral response as shown by others)? (CoV is not a particularly strong trigger of asthma exacerbations). More specifically, did CoV increase IL-25 and trigger the feed forward inflammatory response as modeled by the authors?
5. Figure 5B. To what extent is it possible to measure IL-25 via IHC in the presence of the anti-IL-25 given these mice (the same antibody was used in the therapeutic intervention and the EIA so they can compete)?
6. Figure 5C – these data reflect what dpi? Data in figures 6 and the significant data in day 7 are day 1 so it would be important to examine these results on the same day.
7. Do the authors wish to speculate as to why IL-25 is found at the apex but is thought to act in the submucosa, e.g., its action on pDCs, ILC2s, mast cells, etc. (other than perhaps if it only acts in an autocrine/paracrine fashion).
8. P. 17 there are phase 3 studies for tezepelumab (and it is now FDA approved) and there are now published phase 2 studies for anti-IL-33 (itepekimab) that should be referenced.
9. P. 19. It could be mentioned that along with patchy viral infection, infected cells are exquisitely rapidly cleared including via phagocytosis of apoptotic cells by adjacent epithelial cells.

Reviewer #2 (Remarks to the Author):

The authors have developed an antibody which neutralizes many type 2 effects of RV. They detected colocalization IL-25 and IL-25 receptor at the apical surface of uninfected airway epithelial cells and rhinovirus infection increased IL-25 expression. Analysis of immune transcriptome of rhinovirus-infected differentiated asthmatic bronchial epithelial cells (BECs) treated with an anti-IL-25 monoclonal antibody (LNR125) revealed increased type I/III IFN and reduced expression of type-2 immune genes CCL26, IL1RL1 (ST2/ IL33R) and IL-25 receptor. LNR125 treatment also increased type I/III IFN expression by coronavirus infected BECs. An IL-25 treatment increased viral load with suppressed innate immunity. In vivo LNR125 treatment reduced IL-25/type 2 cytokine expression and increased IFN- β expression and reduced lung viral load.

Much of the data of this work about the effects of IL-25 has been previously published (Sci Transl Med. 2014 Oct 1; 6(256): 256ra134.) by this group in ovalbumin-sensitized mice and submerged cell culture.

The authors suggest "IL-25.. directly inhibits virus induced airway epithelial cell innate anti-viral immunity," but while they present data suggesting IL-25 dependent IFN synthesis suppression,

and hint of an increase in viral replication with IL-25, the data fall short of proving an actual direct mechanistic suppression of innate immunity.

Fig. 1. The authors have previously worked with submerged cell cultures to show RV induces IL-25. The present work is said to be in differentiated cultures, but there can be substantial differences in the degree of differentiation in ALI cultures. There is no estimate of the fraction of differentiated cells, nor is there any criteria like H&E staining of sectioned membranes of ciliated cells, or confocal staining of cilia in membrane whole mounts.

The original biopsies have cilia, but how differentiated are these ALI cells? Cells in culture tend to de-differentiate as they are plated into the well. The authors need an estimate of fractional differentiation given that much of this work depends on comparing the transcriptional response of normal and asthmatic patient-derived differentiated cells.

There is a statement that IL-25 and -IL17RB is expressed at the "apical surface" of differentiated cells. On the cilia? There are no cilia in this image. On the cell surface between cilia? In vesicles in the cytoplasm underneath the cilia? It's impossible to tell from the low magnification image. These could easily be undifferentiated cells between differentiated cells in the culture.

Fig. 2 Much of the work in this paper depends on an antibody to IL-25 the authors have revealed their interest in. The authors show some variance in their antibody's de-repression of RV-stimulated IFN responses with Wilcoxon-signed rank test and Friedmen multiple comparisons test. Much variance seems to remain, however.

Perhaps there is a variance among cultures' differentiation state which could be lowered by comparing cultures with a similar degree of differentiation? Perhaps they need to compare transcriptomes of undifferentiated cells from the two sources as well? One wonders, given the images shown in Fig. 1B, whether the authors even need to bother differentiating the cells to get these effects.

Although this antibody is a monoclonal, there is no detailed characterization of the antibody given. We are given no idea of its specificity. There is no description of the epitope on IL-25 bound, nor are other neutralizing anti-IL-25 antibodies compared to this antibody. Many of the anti-IL-25 monoclonals from other sources are sold as human or mouse-specific. The LNR125 antibody used here appears to work on both human and mouse cells. Why is that, exactly?

Fig. 3 This is an interesting figure again expanding their previously published results in ALI cultures. In Fig. 3 C and D, the authors have reached significance in comparing RV and RV+IL-25 for an increase in copy number/ volume with cytokine treatment with a decrease in IFN b and L copies/volume. It would be more rigorous to compare copies of their targets to transcripts of a housekeeping gene such as glyceraldehyde phosphate dehydrogenase, b-actin, or even 18S RNA. This point is brought home in panels E and F where cytokines are measured by ELISA and variances are much lower. The authors may be able to improve their qPCR by making the proper control normalizations.

Fig. 4 This figure suffers from a similar problem as Fig. 3 C and D. Compare these results to a housekeeping gene to improve interpretability.

Fig, 5 recapitulates their previously published results on the mouse immune response, namely, that blocking IL-25 signalling ablated rhinovirus-exacerbated type-2 leukocytic airways inflammation. In their previous work they blocked the IL-25 receptor. Here, they block IL-25.

Fig. 6 produces the unsurprising result that blocking IL-25 increases IFNs and decreases RV copy number. The direct cytokine measures show a relatively low level of variance compare to the qPCR. These results again need to be calibrated like those in Figs. 3-5. Compare to housekeeping genes, not sample volume.

We wish to thank the editor and reviewers for their time and effort reviewing and improving our manuscript titled "IL-25 blockade augments antiviral immunity during respiratory virus infection" and their detailed and constructive feedback. We provide a detailed point-by-point response to reviewer comments below.

We hope that the revised manuscript will now be considered suitable for publication in *Nature Communications Biology*.

Reviewers' comments:

Reviewer #1 (Remarks to the Author):

This is a well performed study which reports the ability of IL-25 to decrease anti-viral immune responses to RV and as such the ability of an anti-IL-25 antibody to promote viral clearance including through enhanced IFN production and ISG expression and reduced T2 inflammation. Both ex vivo and in an in vivo murine model IL-25 reduced anti-viral immunity and increased viral load and conversely anti-IL-25 restored anti-viral immunity and reduced viral loads. Given the expected limitations of performing research in (outbred) humans, there are only a few minor considerations.

Reviewer Comment

1. IL-17RB expression (figure 1A). It might be expected in the biopsy specimens to see this being expressed on mast cells, ILC2s, eos, and others. No obvious "red" stained cells are observed in this figure.

Response

Thank you for the comment. We agree that IL-17RB expression in endobronchial biopsies is restricted to apical surface of airway epithelium. Given the aim of this part of the study was to demonstrate that airway epithelial cells express both IL-25 and IL-17RB, we focused on tissue sections that predominantly contained intact epithelium. As a result, we would expect the number of immune cells to be relatively low and therefore difficult to identify. Certainly, future analyses could examine biopsies of lung tissue and include CD45 co-staining to identify leucocytes. To acknowledge this point we have added the following text to the revised manuscript discussion (pg. 17):

"The endobronchial biopsies we selected to examine predominantly contained intact airway epithelium and IL-17RB-expressing immune cells were not observed. Future studies could examine biopsies with lung tissue and include CD45 co-staining to identify leucocytes."

Reviewer comment

2. There is a problem with the IL-25 in house EIA insofar as it is using the same antibody for capture and enzyme-linkage, the problem being that the former antibody is likely to stoichiometrically block available epitopes for the latter antibody. This may be contributing to the poor data displayed in 1D (where all the day 4 samples and half the day 2 samples show no increase).

Response

Thank you for the comment. In this case the reviewer is mistaken – I have copied the relevant section from the methods for clarification (pg. 8; bold added for emphasis):

“96 well plates were coated with **LNR126** capture antibody in PBS and stored at 4°C overnight. Plates were washed 3X in PBS and 0.075% Tween-20 and blocked with reagent diluent (R&D systems, USA) at room temperature for one hour. The standard was prepared with recombinant IL-25 (R&D Systems) with a range of 250-1.95 pg/mL in reagent diluent. Samples were diluted in a 1:4 combination of RIPA buffer and reagent diluent. **Biotinylated LNR125** detection antibody was diluted in blocking buffer and incubated for 1 hour at room temperature.”

Please note that a different anti-IL-25 antibody (LNR126, binds to a different epitope to LNR125) was used to capture IL-25. Biotinylated LNR125 was used for detection. We have added additional supplemental data (**supplemental figure 1B**) as evidence that LNR125 and LNR126 each bind to IL-25 without interfering with each other’s binding. We have added additional text to the methods section referring to this supplementary data:

‘Human IL-25 was measured in BEC apical supernatants and cell lysates by sandwich ELISA (Formerly Abeome Corporation, now Lanier Biotherapeutics, United States). This sandwich ELISA format was initially used to perform epitope binning assays to demonstrate that LNR125 does not interfere with LNR126 IL-25 binding and vice versa for human IL-25 ELISA assays (**Supplementary Figure 1B**)’.

Reviewer comment

3. *Figure 2A is impressive. Figure 2 is missing a label for “D”.*

Response

Thank you for noting this – the label for panel D has been added to the updated figure.

Reviewer comment

4. *The authors should discuss more reasons as to why the response to RV and CoV are so different. Perhaps this points to the more T2-inducing effects of RV that are being abrogated by the anti-IL-25 antibody. Did the CoV have a similar impact on the T2 cytokines (possibly with the high T2 response to RV triggering an anti-viral response as shown by others)? (CoV is not a particularly strong trigger of asthma exacerbations). More specifically, **did CoV increase IL-25 and trigger the feed forward inflammatory response as modeled by the authors?***

Thank you for this excellent suggestion. We have measured expression of IL-25 mRNA and protein in BECs infected with CoV (without antibody treatment as this would potentially interfere with IL-25 protein detection). Unlike infection with RV, 229E did not induce increased expression of IL-25 mRNA and protein (new Figure 4E and F). Although this is consistent with the idea that RV is a stronger (than 229E) trigger of asthma exacerbations by stimulating IL-25 this must be considered with caution. The data for RV-induced expression is derived from BECs from 14 donors at two time points (2 and 4 days p.i.). Although

statistically significant, induction was heterogenous – with some donors’ BECs not exhibiting increased IL-25 with RV infection. For 229E, we used expanded cells from two donors and examined induction at single time point (3 days post-infection). Again, viral induction of IL-25 was heterogenous and was not significant. It is possible that if we examined cells from more donors and additional timepoints following 229E infection we may detect increased expression of IL-25. To acknowledge this, we have added the following points to the discussion (pg. 21):

“We also examined induction of IL-25 and effects on anti-viral immunity during CoV 229E infection. Unlike RV, we did not observe 229E-increased IL-25 expression by differentiated primary human BECs, although blocking IL-25 did increase virus-induced IFN- λ . This observation is consistent with RV having greater capacity to promote type 2 inflammation and might underly the greater prevalence of this virus in triggering asthma exacerbations. However, the RV-induced IL-25 and 229E induced IL-25 datasets are not directly comparable: the data for RV-induced IL-25 was derived from BECs from 14 donors and assessed at two time points (2 and 4 days post infection). Although statistically significant RV-induced IL-25 expression, this was heterogenous with some donors’ BECs not exhibiting increased IL-25 with RV infection. For 229E, we used BECs from two donors (replicates for each donor) and examined induction at single time point (3 days post-infection) and did not observe viral increased IL-25 expression. It is possible that if we examined cells from more donors and additional timepoints following 229E infection we would detect increased expression of IL-25 indicating that 229E, like RV, can stimulate type 2 inflammation and cause asthma exacerbation. Further studies that directly compare virus-induced IL-25 in BECs from the same donors are required to formally address this.”

Additional data has been included in Figure 4, as follows and noted in the revised manuscript results (pg. 14):

Reviewer comment

5. Figure 5B. To what extent is it possible to measure IL-25 via IHC in the presence of the anti-IL-25 given these mice (the same antibody was used in the therapeutic intervention and the EIA so they can compete)?

Response

Excellent question – we were also wary of this being an issue. Due to this very concern of antibody interference affecting any IHC using our anti-IL-25 antibodies, we opted for a commercial mouse DuoSet ELISA (R&D Systems) for assessing lung IL-25 protein. These kits use different antibodies from the antibody administered therapeutically, so this assay should not be interfered with by LNR125 in the lung of treated mice. Our decision was also guided by the ability to directly quantify total lung IL-25 protein by ELISA.

Reviewer comment

6. Figure 5C – these data reflect what dpi? Data in figures 6 and the significant data in day 7 are day 1 so it would be important to examine these results on the same day.

Response

Thank you for the comment – We believe you are referring to the data presented in figures 6 and significant data in FIGURE 7 are day 1 – this is correct. The data shown in Figure 5C is also day 1 (the figure legend has been amended in the revised manuscript to clarify this point)

Type 2 cytokine expression is resolved before day 7, so the Th2 cytokine data in Figure 5C and the IFN data in Figure 6 are from day 1 post-infection. The figure legend for Figure 5C has been updated to indicate the timepoint of this data.

Reviewer comment

7. Do the authors wish to speculate as to why IL-25 is found at the apex but is thought to act in the submucosa, e.g., its action on pDCs, ILC2s, mast cells, etc. (other than perhaps if it only acts in an autocrine/paracrine fashion).

Response

Excellent question. We suspect that some IL-25 protein is trapped in the mucus layer and binds to apical IL-17RB, which enables IL-25 protein to be observed at the apical surface. IL-25 protein secreted baso-laterally would be expected to diffuse into the submucosa extra-cellular matrix and be available to stimulate the immune cells migrating between circulation and airways as you suggest.

Reviewer comment

8. P. 17 there are phase 3 studies for tezepelumab (and it is now FDA approved) and there are now published phase 2 studies for anti-IL-33 (itepekimab) that should be referenced.

Response

Thank you for pointing out these recent clinical studies. We have been anticipating release of data from these studies for some time! We have updated the discussion accordingly (pg. 19):

“Both anti-TLSP (tezepelumab) and anti-IL-33 (itepekimab) has reported phase 2 clinical data, anti-TLSP showing reduction in frequency of asthma exacerbations¹³ and anti-IL-33 provided some protection against loss of asthma control⁵⁷”

Reviewer comment

9. P. 19. It could be mentioned that along with patchy viral infection, infected cells are exquisitely rapidly cleared including via phagocytosis of apoptotic cells by adjacent epithelial cells.

Response

Excellent point – and indeed we see this very clearly in our ALI-BEC cultures: RV harbouring cells are very efficiently eliminated, presumably as you suggest via phagocytosis (efferocytosis) or extrusion. We have addressed this point in the revised discussion (pg. 20):

“Infected cells are eliminated by a process of extrusion (ejected from the epithelium) or epithelial cell phagocytosis (efferocytosis)⁶⁰”

Reviewer #2 (Remarks to the Author):

The authors have developed an antibody which neutralizes many type 2 effects of RV. They detected colocalization IL-25 and IL-25 receptor at the apical surface of uninfected airway epithelial cells and rhinovirus infection increased IL-25 expression. Analysis of immune transcriptome of rhinovirus-infected differentiated asthmatic bronchial epithelial cells (BECs) treated with an anti-IL-25 monoclonal antibody (LNR125) revealed increased type I/III IFN and reduced expression of type-2 immune genes CCL26, IL1RL1 (ST2/ IL33R) and IL-25 receptor. LNR125 treatment also increased type I/III IFN expression by coronavirus infected BECs. An IL-25 treatment increased viral load with suppressed innate immunity. In vivo LNR125 treatment reduced IL-25/type 2 cytokine expression and increased IFN- β expression and reduced lung viral load.

Reviewer comment

Much of the data of this work about the effects of IL-25 has been previously published (Sci Transl Med. 2014 Oct 1; 6(256): 256ra134.) by this group in ovalbumin-sensitized mice and submerged cell culture.

Response

Thank you for the comment. We do agree that some of the published data does confirm the pivotal role of IL-25 in mediating the virus-increased allergic airways inflammation. The key advance in the current manuscript is to show that this can be achieved by targeting the cytokine directly (rather than IL-25 receptor as published previously). We believe that this point of difference has significant implications for clinical development of anti-IL-25 biologics.

Reviewer comment

The authors suggest "IL-25 directly inhibits virus induced airway epithelial cell innate anti-viral immunity," but while they present data suggesting IL-25 dependent IFN synthesis suppression, and hint of an increase in viral replication with IL-25, the data fall short of proving an actual direct mechanistic suppression of innate immunity.

Response

Thank you for the comment and you are right to point this out. Nanostring mRNA expression data suggested that IL-25 regulates expression of key molecular components of the anti-viral signalling pathway including IRF7 and TBK-1. We did attempt to confirm this the protein level, however were unable to demonstrate an effect. To acknowledge this the following statement has been added to the revised manuscript discussion (pg. 18):

"The precise mechanism by which IL-25 inhibits anti-viral innate immunity remains to be elucidated. Nanostring mRNA expression analysis suggested up-regulation of key molecules in anti-viral signalling pathway including TBK-1, IRAK2 and IRF7. However, we were unable to confirm this at the protein level due to heterogeneity of donor BEC responses. IL-25 signalling can activate multiple transcription factors including p38 mitogen activated kinase, c-Jun, NF- κ B and STAT5^{43, 44}. Of these, STAT5 is a candidate for interfering with STAT1 mediated interferon responses⁴⁵. "

Reviewer comment

Fig. 1. The authors have previously worked with submerged cell cultures to show RV induces IL-25. The present work is said to be in differentiated cultures, but there can be substantial differences in the degree of differentiation in ALI cultures. There is no estimate of the fraction of differentiated cells, nor is there any criteria like H&E staining of sectioned membranes of ciliated cells, or confocal staining of cilia in membrane whole mounts. The original biopsies have cilia, but how differentiated are these ALI cells? Cells in culture tend to de-differentiate as they are plated into the well. The authors need an estimate of fractional differentiation given that much of this work depends on comparing the transcriptional response of normal and asthmatic patient-derived differentiated cells. There is a statement that IL-25 and -IL17RB is expressed at the "apical surface" of differentiated cells. On the cilia? There are no cilia in this image. On the cell surface between cilia? In vesicles in the cytoplasm underneath the cilia? It's impossible to tell from the low

magnification image. These could easily be undifferentiated cells between differentiated cells in the culture.

Response

Thank you the comment – we absolutely agree that confirming the differentiation status of ALI cultures is critical. We have an established track record of generating well-differentiated (ALI) primary human airway epithelial cell cultures and have published numerous papers that provide histological evidence that cultures are fully differentiated. These include (but not limited to):

- Reid AT, Nichol KS, Veerati PC, Moheimani F, Kicic A, Stick SM, Bartlett NW, et al., 'Blocking notch3 signaling abolishes MUC5AC production in airway epithelial cells from individuals with asthma', *American Journal of Respiratory Cell and Molecular Biology*, 62 513-523 (2020)
- Veerati PC, Troy NM, Reid AT, Li NF, Nichol KS, Kaur P,... Bartlett NW, et al., 'Airway Epithelial Cell Immunity Is Delayed During Rhinovirus Infection in Asthma and COPD', *Frontiers in Immunology*, 11 (2020)
- Singanayagam A, Glanville N, Girkin JL, Ching YM, Marcellini A, Porter JD, Bartlett NW, et al., 'Corticosteroid suppression of antiviral immunity increases bacterial loads and mucus production in COPD exacerbations', *NATURE COMMUNICATIONS*, 9(2229) (2018)

We are not clear on how to calculate 'fractional differentiation' - we routinely culture primary human airway epithelial cells at ALI for at least 25 days and monitor transepithelial electrical resistance, mucous production and cilia activity. Only cultures that exhibit sufficiently high TEER, produce mucous and have clearly observable beating cilia by light microscopy are used for studies. As evidence of this, the ALI-BEC sections used for immunofluorescence detection of IL-25 and IL-17RB shown in figure 1B were derived from blocks generated from ALI-BEC cultures originally described in Veerati PC *et al* 2020. In that paper, we provide histology and mucin immunostaining data to confirm differentiation (Figure 3C from Veerati *et al* 2020 included below), which exhibits stratified epithelium (consistent with the DAPI nuclear staining presented in Figure 1B) with cilia and mucin (MUC5AC) positive goblet cells present.

c MUC5AC IHC – 96 hpi

We have added the following statement to the methods section to address this point (pg. 9):

“We have previously confirmed the differentiation status (stratified, active cilia, mucin-positive goblet cells) of the ALI-BEC cultures from which sections were subsequently obtained for immunofluorescence for the current study²⁵. “

We have also included the following in the revised results section (pg. 11):

“We next determined if ALI-differentiated BEC cultures from healthy and asthmatic donors generated for a previous study²⁵ exhibited a similar IL-25 and IL-17RB expression pattern to that of bronchial biopsies. By comparing the pattern of expression to previously reported histological analyses of these ALI-BEC cultures, we again observed that IL-25 and IL-17RB expression was highly localized to the apical surface.”

Reviewer comment

Fig. 2 Much of the work in this paper depends on an antibody to IL-25 the authors have revealed their interest in. The authors show some variance in their antibody's de-repression of RV-stimulated IFN responses with Wilcoxon-signed rank test and Friedmen multiple comparisons test. Much variance seems to remain, however.

Perhaps there is a variance among cultures' differentiation state which could be lowered by comparing cultures with a similar degree of differentiation? Perhaps they need to compare transcriptomes of undifferentiated cells from the two sources as well? One wonders, given the images shown in Fig. 1B, whether the authors even need to bother differentiating the cells to get these effects.

Response

Thank you for the comment. We believe that using well-differentiated airway epithelial cell cultures derived from multiple human donors provides the strongest evidence to support our conclusions. A challenge with using this physiologically informative model is donor to donor variability, as highlighted by the reviewer. However, there is no question that that the cultures are differentiated (addressed above) and therefore we do not see the value in undertaking extensive additional transcriptomic analyses in undifferentiated cells.

Reviewer comment

Although this antibody is a monoclonal, there is no detailed characterization of the antibody given. We are given no idea of its specificity. There is no description of the epitope on IL-25 bound, nor are other neutralizing anti-IL-25 antibodies compared to this antibody. Many of the anti-IL-25 monoclonals from other sources are sold as human or mouse-specific. The LNR125 antibody used here appears to work on both human and mouse cells. Why is that, exactly?

Response

Thank you for the comment – we have provided additional information on how the antibody was sourced/generated, including the species specificity and epitope-mapping. This information has been added to the revised methods section (pg. 5):

“Anti-IL-25 antibody LNR125 and LNR126 (developed by Abeome Corporation, now Lanier Biotherapeutics, Athens, GA, USA) was generated using the AbeoMouse™ platform which allows selection of affinity matured B cells via cell surface antibody selection (**Supplementary Figure 1A**). Neutralising potency of anti-IL-25 antibodies against mammalian cell expressed human- and mouse IL-25 was determined by calculating IC₅₀ in HT29 cell reporter assay and indicated that LNR125 neutralised both human and mouse whereas LNR126 neutralised human IL-25 only. A sandwich ELISA format was used to perform epitope binning assays and determine that LNR125 does not interfere with LNR126 IL-25 binding and vice versa and therefore this antibody pair was identified as suitable for human IL-25 ELISA development (**Supplementary Figure 1B**).”

Reviewer comment

Fig. 3 This is an interesting figure again expanding their previously published results in ALI cultures. In Fig. 3 C and D, the authors have reached significance in comparing RV and RV+IL-25 for an increase in copy number/ volume with cytokine treatment with a decrease in IFN β and L copies/volume. It would be more rigorous to compare copies of their targets to transcripts of a housekeeping gene such as glyceraldehyde phosphate dehydrogenase, b-actin, or even 18S RNA. This point is brought home in panels E and F where cytokines are measured by ELISA and variances are much lower. The authors may be able to improve their qPCR by making the proper control normalizations.

Response

We thank the reviewer for their advice on how to improve the qPCR data and we apologise for any misunderstanding. The y-axis states the copy number per μ L of cDNA purely for the ease of interpretation by readers from the wider community. The qPCR is normalised to 18s house keeper gene expression, as detailed in the methods (pg. 7):

“Gene copy numbers were normalized to housekeeping gene 18S and quantitated using a standard curve”

Reviewer comment

Fig. 4 This figure suffers from a similar problem as Fig. 3 C and D. Compare these results to a housekeeping gene to improve interpretability.

Response

In this case, reviewer 2 is mistaken. Figure 4A is normalised as per the response above. Figure 4B-D are ELISA data, so housekeeper gene normalisation is not applicable for this data.

Reviewer Comment

Fig. 5 recapitulates their previously published results on the mouse immune response, namely, that blocking IL-25 signalling ablated rhinovirus-exacerbated type-2 leukocytic airways inflammation. In their previous work they blocked the IL-25 receptor. Here, they block IL-25.

Response

We thank the reviewer for their acknowledgment of our previous work and raising this key point. We agree that the data is in line with the previous work, and that we can now directly confirm a role for IL-25 (IL-17E), which we could not previously do given that both IL-17B and IL-17E utilise IL-17RB. The fact that we can confirm the role of IL-25 as a central mediator in viral increased airway inflammation is a key finding needed to support clinical development of LNR125. To clarify this, we have added the following statement to the revised discussion (pg. 22):

“Two IL-17 family cytokines signal via IL-17RB – IL-17E (IL-25) and IL-17B. Therefore, blocking IL-17RB was not definitive proof that IL-25 was the cytokine driving lung type 2 inflammation, although it was highly likely given lung expression and established role in type 2 immunity for IL-25¹⁴ versus low lung expression and distinct biological activity for IL-17B⁶⁷. Using a mouse model of RV-exacerbation of allergic airway disease, we found LNR125 reduced total BAL cell counts, and secretion of IL-4, IL-5, and IL-25. This confirmed a central role for IL-25 in regulating virus induce airway inflammation.”

Reviewer comment

Fig. 6 produces the unsurprising result that blocking IL-25 increases IFNs and decreases RV copy number. The direct cytokine measures show a relatively low level of variance compare to the qPCR. These results again need to be calibrated like those in Figs. 3-5. Compare to housekeeping genes, not sample volume.

Response

We have clarified the use of housekeeping gene normalisation for qPCR analyses above.

REVIEWERS' COMMENTS:

Reviewer #1 (Remarks to the Author):

I am very satisfied with the thorough response to my previous concerns. This is a series of neat observations.

Reviewer #2 (Remarks to the Author):

The comments that this reviewer raised have been answered.